# Lysyl Oxidase-Like 2 Protects against Progressive and Aging Related Knee Joint Osteoarthritis in Mice

**DOI:** 10.3390/ijms20194798

**Published:** 2019-09-27

**Authors:** Mustafa Tashkandi, Faiza Ali, Saqer Alsaqer, Thabet Alhousami, Amparo Cano, Alberto Martin, Fernando Salvador, Francisco Portillo, Louis C. Gerstenfeld, Mary B. Goldring, Manish V. Bais

**Affiliations:** 1Department of Molecular and Cell Biology, Boston University Henry M. Goldman School of Dental Medicine, Boston, MA 02118, USA; mtash@bu.edu (M.T.); fali7@bu.edu (F.A.); Alsaqer@bu.edu (S.A.); thabet@bu.edu (T.A.); 2Department of Periodontology, Boston University Henry M. Goldman School of Dental Medicine, Boston, MA 02118, USA; 3Departamento de Bioquímica, Universidad Autónoma de Madrid, Instituto de Investigaciones Biomédicas “Alberto Sols” CSIC-UAM, IdiPAZ, 28029 Madrid, Spain; acano@iib.uam.es (A.C.); almmartin@isciii.es (A.M.); fernando.salvador@irbbarcelona.org (F.S.); fportillo@iib.uam.es (F.P.); 4Centro de Investigación Biomédica en Red Cáncer. Av Monforte de Lemos, 3-5, Pabellón 11, planta 0, 28029 Madrid, Spain; 5Department of Orthopedic Surgery, School of Medicine, Boston University, Boston, MA 02118, USA; lgersten@bu.edu; 6Hospital for Special Surgery Research Institute, and Department of Cell and Developmental Biology, Weill Cornell Medical College, New York, NY 10021, USA; mbgoldring@hss.edu

**Keywords:** Lysyl oxidase like-2, adenovirus delivery, knee joint, articular cartilage, regeneration, osteoarthritis, anabolic response

## Abstract

Background: The goal of this study was to determine if adenovirus-delivered LOXL2 protects against progressive knee osteoarthritis (OA), assess its specific mechanism of action; and determine if the overexpression of LOXL2 in transgenic mice can protect against the development of OA-related cartilage damage and joint disability. Methods: Four-month-old Cho/+ male and female mice were intraperitoneally injected with either Adv-RFP-LOXL2 or an empty vector twice a month for four months. The proteoglycan levels and the expression of anabolic and catabolic genes were examined by immunostaining and qRT-PCR. The effect of LOXL2 expression on signaling was tested via the pro-inflammatory cytokine IL1β in the cartilage cell line ATDC5. Finally; the OA by monosodium iodoacetate (MIA) injection was also induced in transgenic mice with systemic overexpression of LOXL2 and examined gene expression and joint function by treadmill tests and assessment of allodynia. Results: The adenovirus treatment upregulated LOXL2; Sox9; Acan and Runx2 expression in both males and females. The Adv-RFP-LOXL2 injection; but not the empty vector injection increased proteoglycan staining and aggrecan expression but reduced MMP13 expression. LOXL2 attenuated IL-1β-induced phospho-NF-κB/p65 and rescued chondrogenic lineage-related genes in ATDC5 cells; demonstrating one potential protective mechanism. LOXL2 attenuated phospho-NF-κB independent of its enzymatic activity. Finally; LOXL2-overexpressing transgenic mice were protected from MIA-induced OA-related functional changes; including the time and distance traveled on the treadmill and allodynia. Conclusion: Our study demonstrates that systemic LOXL2 adenovirus or LOXL2 genetic overexpression in mice can protect against OA. These findings demonstrate the potential for LOXL2 gene therapy for knee-OA clinical treatment in the future.

## 1. Introduction

The progressive joint disease osteoarthritis (OA) affects a significant portion of the US population but has a few therapeutic options. The estimated lifetime risk of knee OA is 14% [1], with annual healthcare costs exceeding $185 billion in the United States [2], and the prevalence of OA is increasing [3]. However, no anabolic agent has been approved for clinical application.

Using transcriptomics in a mouse model, the authors previously demonstrated that copper-dependent amine oxidase lysyl oxidase-like 2 (LOXL2) is specifically upregulated during fracture healing and mediates endochondral ossification [4]. Our follow-up study also showed that LOXL2 is a critical regulator of the chondrogenic lineage [5]. Further, LOXL2 also was found to have a novel anabolic function in OA cartilage [6,7]. The microarray analysis demonstrates that the transduction of human OA articular chondrocytes with Adv-LOXL2 increases the expression of anabolic genes [6]. Thus, LOXL2 is a mediator of both the chondrogenic lineage and cartilagenous changes affected by OA and thus may have therapeutic potential. 

Our previous data inspired the following questions. (1) Can LOXL2 attenuate progressive knee-OA-related degenerative changes? (2) If LOXL2 exerts a protective function, can it be exploited in pre-clinical models to uncover new therapeutic strategies for OA? 

The recent advances and success of gene therapy for delivering therapeutic molecules [8] has stimulated interest in this field, and the application of adenovirus and lentivirus by direct intra-articular joint and intraperitoneal injections in preclinical and clinical studies is now trending [9]. The studies show that adenovirus-mediated administration of bFGF, IL-1Ra, or IGF-1 is effective in a rat OA model [10]. Therefore, this study aimed to determine if adenoviral LOXL2 acts via an anabolic mechanism to retard the progressive knee OA and if the overexpression of LOXL2 protects against OA-related cartilage damage and joint dysfunction. Considering the benefits of the adenovirus delivery system [11], this study used this system to deliver *LOXL2* into an in vitro OA model or, using the systemic injection, into knee joint OA mouse models. 

The three models used to uncover the chondroprotective effect of LOXL2 are: Chondrodysplasia (Cho/+) mice, a cartilage cell line exposed to the inflammatory cytokine IL-1β, and monosodium iodoacetate (MIA)-induced LOXL2 transgenic mice. The Cho/+ mouse, which carries a single-nucleotide deletion resulting in the premature termination of the α1 chain of type XI collagen [12], is a good model to study the progressive OA changes that develop with aging. The type XI collagen expression is reduced in the articular cartilage of older individuals [13], and the data from human and animal experiments show that the decreased type XI collagen levels in cartilage might be one initiating factor of tempoormandibular joint [14] and knee joint-OA pathogenesis [15]. Next, in vitro IL-1β-induced catabolic changes and molecular signaling mechanisms were assessed in a cartilage cell line. Of note, the IL-1β inhibitors have been implicated as a therapy for OA [16,17]. Finally, this study tested whether the endogenous overexpression of LOXL2 can protect against MIA-induced knee OA and alleviate OA-related structural and functional consequences, including the proteoglycan and aggrecan expression, joint mobility, and allodynia. The MIA-induced model is extensively used for knee OA-related studies in both mice and rats. This model allowed the assessment of the efficacy of the genetic overexpression of LOXL2 in attenuating OA-related functions. 

## 2. Results

### 2.1. Adenoviral LOXL2 Protects Against Progressive OA in Cho/+ Mice

To test whether LOXL2 induces anabolic responses in progressive mouse knee-OA, Cho/+ mice injected with Adv-RFP-LOXL2 were compared to mice injected with Adv-RFP-Empty. First, this study evaluated if the adenovirus vector induced any adverse effects, such as the cytotoxicity or immune reaction in the mice. The data showed that the adenovirus vector injection does not have any adverse effects on the total LDH release of ATDC5 cells. Next, serum samples from mice injected with vehicle or the adenovirus vector (Adv-RFP-Empty) was evaluated and found to have no significant differences in serum total LDH (Appendix A). The evaluation of tissues for the expression of inflammatory cytokines, such as TNFα and IL6, does not show any effect due to the adenovirus vector injection compared to the vehicle injected group. As shown in Figure 1A, the equal numbers of male and female Cho/+ mice were injected intraperitoneally with either Adv-RFP-Empty or Adv-RFP-LOXL2 twice monthly for 12 weeks. Safranin-O/Fast Green staining showed significantly increased proteoglycan deposition (Figure 1B). Immunostaining revealed that Adv-RFP-LOXL2 increased the expression of LOXL2 aggrecan and Col2 but reduced expression of Mmp13 and Adamts5 (Figure 1C–I). The knee joints from both male and female mice injected with Adv-RFP-LOXL2 showed the higher expression of *LOXL2*, *Acan*, *Sox9*, *Col2a1,* and *Runx2* than knee joints from mice injected with Adv-RFP-Empty (Figure 2). The data also shows that TNFα and IL6 expression was not affected significantly. The forced expression of LOXL2 significantly increased mRNA levels of anabolic genes, indicating that LOXL2 promotes anabolism in general. For both male and female mice, the expression of *Vegf-b* and *Col10* were not affected by LOXL2-forced expression. However, LOXL2-forced expression caused a non-significant increase in *Mmp13* and *Rankl* in females only (Figure 2). Overall, these data suggest that the expression of LOXL2 protects against proteoglycan loss and OA progression and may stimulate a protective response in knee joint articular cartilage via the factors secreted in the OA joint.

### 2.2. LOXL2 Attenuates IL-1β Induced NF-κB in Cartilage

IL-1β is a pro-inflammatory factor secreted in the knee joint and promotes catabolism. To test if LOXL2 acts partly by attenuating IL-1β-induced catabolic responses, ATDC5 cartilage cells were treated with IL-1β alone or in combination with Adv-RFP-LOXL2, and molecular signaling and gene expression were assessed (Figure 3). IL-1β reduced mRNA levels of *Acan* and *Sox9*, but LOXL2 blunted this effect. Finally, LOXL2 attenuated the expression of inducers of IL-1β, including Admts4/5 and MMP13 after 1 and/or 3 days of treatment (Figure 3). Thus, LOXL2 may protect against pro-inflammatory signaling mechanisms and restore the expression of cartilage-related genes.

### 2.3. Enzymatically Inactive LOXL2 also Attenuates IL-1β-Induced NF-κB Activity

IL-1β induced phosphorylation of NF-κB/p65, but this was attenuated by the overexpression of LOXL2 (Figure 4A,B). To test if the anti-catabolic effect of LOXL2 is mediated by an enzymatic or non-enzymatic mechanism, ATDC5 cells were treated with IL-1β and Adv-LOXL2 in the presence or absence of the LOX family inhibitor β-aminoproprinytryl (BAPN). The abilitry of LOXL2 expression to block NK-κB phosporylation was not prevented by BAPN (Figure 4A,B). These data indicate that the ability of LOXL2 to inhibit OA-related structural and functional changes could be indepenent of its enzymatic activity. 

### 2.4. LOXL2 Overexpression Protects Against Catabolic Changes in Aging Knee OA

To evaluate if endogenous LOXL2 can alleviate knee-OA-related catabolic changes and functional consequences in aging mice, 14-month-old LOXL2 homozygous transgenic mice intra-articularly injected with MIA were used to assess histological and functional changes (Figure 5A). The mice overexpressing LOXL2 were engineered and validated by Cano laboratory [18]. Safranin-O staining showed that proteoglycan expression was similar in untreated LOXL2 and wild-type (WT) littermates, but proteoglycan was depleted in knee joints injected with MIA in WT mice. This depletion was ameliorated in LOXL2 transgenic mice (Figure 5B). Similar to LOXL2 expression (Figure 5C), the Acan expression appeared to be higher in untreated LOXL2 transgenic and WT littermates, but decreased in knee joints of injected mice with MIA (Figure 5D). Again, LOXL2 overexpression in the transgenic mice prevented this depletion. Thus, LOXL2 protects against MIA-induced proteoglycan and aggrecan degradation as well as decreased Mmp13 expression in knee OA.

### 2.5. Overexpression of LOXL2 Protects Against Functional Changes in Aging Knee OA 

MIA-induced OA produces pain-depressed wheel running [19]. Treadmill behavior is useful for preclinical behavioral assessment of chronic pain and inflammation [20]. These LOXL2 transgenic mice were used to evaluate if the constitutive overexpression of LOXL2 protects against OA pain and inflammation. The treadmill analysis was performed with a standard protocol at 28 days post-MIA injection. The LOXL2-overexpressing mice ran longer and farther than wild-type littermates (Figure 6A,B). The LOXL2-overexpressing mice also exhibited maximal exercise capacity and improved allodynia compared to WT littermates (Figure 6C). Thus, LOXL2 appears to protect mice from MIA-induced OA and restore joint function.

## 3. Discussion

While seeking to understand how LOXL2 attenuates catabolic factors during age-related OA pathogenesis, the overarching goal was to evaluate if adenoviral LOXL2 could be used for translational research and future clinical applications in OA treatment. In a prior study, the authors showed that LOXL2 could have chondroprotective and anabolic effects in OA [6], but the in vivo role of LOXL2 in cartilage development, maintenance, and protection from OA is not known. The data from our current study indicate that LOXL2 could induce a chondroprotective response by inhibiting catabolic factors or IL-1β-induced NF-κB signaling pathways. 

First, this study utilized 4-month-old Cho/+ mice, which developed progressive knee-OA [14,15,21,22], to evaluate the protective effect of LOXL2 and understand the regulation of key regulatory targets involved in knee-OA. The knee joints of 3-month old Cho/+ mice were normal but exhibited progressive OA-like changes with age-associated increases in severity. By 6 months of age, the proteoglycan staining was lost in superficial zones, and by 9 months, the region of deficient proteoglycan staining extended from superficial to deep layers. By 15 months of age, the typical OA-like knee joints, including the loss of articular cartilage, misshaped meniscus, and the inflammation in synovial tissues, have developed in Cho/+ mice, but not WT littermates [15]. At 3 and 6 months, the expression of ECM-degrading proteins MMP-3 and MMP-13 were elevated, respectively, relative to WT levels in knee joints of Cho/+ mice. MMP-13 is implicated in OA pathogenesis [23], and our current study shows that LOXL2 can attenuate MMP-13 expression in vitro and in vivo. 

Next, the above histological and molecular findings were validated using an additional mouse model and extended our study to examine joint function. The LOXL2-overexpressing transgenic mice were more resistant to MIA-induced OA, indicating that LOXL2 has a chondroprotective function and can inhibit disability as evaluated by the treadmill test and allodynia. A previous study showed that LOXL2-induced collagen cross-linking enhances the tensile strength of articular cartilage and the resistance to collagen proteolysis [24]. However, that study did not evaluate the specific mechanism underlying that behavior.

This study sought to evaluate a possible mechanism by examining the effect of the LOXL2 expression on the catabolic IL-1β-induced signaling pathway. Of note, IL-1β is a target of various therapeutic interventions. In chondrocytes, IL-1β activates the canonical NF-κB pathway, which is dependent upon IKKβ, leading to increased gene expression of MMPs, ADAMTSs, inflammatory mediators (COX2, NO, PGE2), chemokines (IL-8), and cytokines (IL-1β, TNF-α, IL-6) involved in cartilage destruction. In contrast, the non-canonical NF-κB activating kinase promotes hypertrophy [25]. Inhibiting the IL-1β-induced NF-κB pathway can attenuate OA [16,17], and IL-1β promotes ADAMT5 and MMP13 in OA. It has been determined that LOXL2 inhibits IL-1β-induced NF-κB and catabolic mediator pathways, which are potential mechanisms for progressive OA and aging cartilage.

Although our data show that LOXL2 has a protective function in knee OA, many questions remain. It is unknown whether defective LOXL2 function leads to OA. The mutation of collagen XI (Cho/+) [15] or *Col2a1* [26] in mice disrupts the cartilage matrix, increasing susceptibility to degradation [27]. However, it is not known if there is a direct correlation between functional dysregulation of LOXL2 and human OA, which needs to be evaluated in the future. Of note, our study shows that the enzymatic inhibition of LOXL2 does not block its ability to attenuate NF-κB activity, demonstrating that its anti-catabolic effect could be mediated by a non-enzymatic region of LOXL2. This needs to be further evaluated, but is in line with other LOXL2 actions previously reported as independent of its catalytic activity [28,29].

With the initial success of gene therapy, the development of specific gene therapy approaches for several diseases and conditions is trending. Our study showed that adenovirus LOXL2 delivery at a specific concentration protected against knee-OA with no adverse effects. However, there is a need to establish a dose-response and safe delivery methods for clinical application. An earlier study showed that LOXL2 overexpressing mice does not develop cancer, however, the application of carcinogen could promote cancer growth or metastasis in breast cancer models [30]. Our study evaluated the systemic delivery of adenovirus LOXL2. However, local delivery could be tested in preclinical and clinical studies in the future. The understanding of detailed molecular and cellular mechanisms could be very promising for future applications. Thus, this study provides a foundation for several future mechanistic and translational studies.

In conclusion, our studies found that: 1) Adenoviral LOXL2 attenuates the effect of OA-related catabolic factors released during progressive OA in the Cho/+ mouse model; 2) the constitutive expression of LOXL2 has no adverse effects and attenuates knee-OA-related changes and preserves normal joint function in aged mice; 3) LOXL2 attenuates IL-1β-induced signaling, which could be a potential mechanism for its protective effect(s). This study demonstrates the importance of developing a more detailed understanding of a LOXL2 function and that adenovirus LOXL2 is a candidate for future translational applications.

## 4. Materials and Methods

### 4.1. Animal Experiments

All mouse experiments were performed with the guidance, regulation, and approval of the Boston University Institutional Animal Care and Use Committee (IACUC; approval number AN-15387, dated Aug 24, 2016). The animal study conformed to ARRIVE guidelines.

### 4.2. Preparation of Adenovirus

Adenoviruses for LOXL2 expression (Ad-CMV-RFP-CMV-hLOXL2-His, referred to as Ad-RFP-LOXL2) and empty vector control (Ad-RFP-Empty) were custom synthesized (ADV-214438) by Vector Biolabs. The adenovirus particles were amplified in 293T cells and quantified by using an adenovirus quantification kit (Cell Biolabs, Inc., San Diego, CA, USA). 

### 4.3. Cho/+ Mouse Experiments

The Cho/+ mice were obtained from Dr. Yefu Li at Harvard Medical School and bred with C57/BL6 mice to maintain a heterozygous mouse colony. The mice were genotyped by PCR using standard protocols. The sixteen-week-old Cho/+ mice were divided into 2 groups: Adv-RFP-Empty was administered to 14 male and 14 female mice, and Adv-RFP-LOXL2 was administered to 14 male and 14 female mice via twice monthly intraperitoneal injection (100 µL; 10^13^ infectious particles/mL) for 3 months. The knee joints were harvested and processed for RT-qPCR and histology.

### 4.4. Cytotoxicity and Immune Effect of Adenovirus Vector

The in vitro cytotoxicity was evaluated in ATDC5 cells by treating the cells with vehicle and adenovirus vector (Adv-RFP-Empty) by using Lactate dehydrogenase (LDH) assay as per the manufacturers’ instruction (Abcam Inc, Cambridge, MA, USA, ab102526). The serum samples of the mice injected with vehicle or adenovirus vector (Adv-RFP-Empty) were dilated 1:10 and 50 uL of serum samples may be evaluated for LDH assay (Abcam, Inc.). In order to evaluate the effect of the adenovirus vector on cytokine expression, RNA from the knee joint from the mice injected with vehicle or adenovirus vector (Adv-RFP-Empty) were subjected to RT-qPCR analysis (Taqman gene expression assay, Applied Biosystems, Waltham, MA, USA) for the expression of TNFα and IL6.

### 4.5. RNA Isolation and Analysis

The total RNA was extracted by Trizol and RNAeasy protocol according to the manufacturer’s instructions (Qiagen, Germantown MD, USA). The quantitative real-time PCR (RT-qPCR) analysis was performed using TaqMan gene expression assays from Life Technologies, according to a standard protocol [31].

### 4.6. Histology and Immunostaining

The knee joints from Cho/+ mice or LOXL2 transgenic mice were paraffin embedded, decalcified, and subjected to histological analysis and immunostaining. Safranin-O/Fast green (American Mastertek Inc., Lodi, CA, USA) staining was performed as described [6]. The OARSI scoring was performed as per the recommendations. Three sections from 4 mice/ group/condition were de-paraffinized and immunostained with specific antibodies to detect RFP, LOXL2, aggrecan, and MMP13 (Abcam) and visualized with HRP-linked anti-rabbit antibodies. The stained tissues were scanned with a digital slide scanner (panoramic MIDI, 3D Histech, Budapest, Hungary).

### 4.7. MIA Mouse Model and Treadmill Running in Aging Mice

The mice with a conditionally targeted *Loxl2* transgene (loxP-PGK-neo-stop-loxP-LOXL2-IRES-eGFP) were crossed to ROSA26-Cre to express LOXL2 constitutively, as shown in our collaborator’s study [18]. To evaluate if the constitutive overexpression of LOXL2 in mice protects from OA pain and inflammation, MIA (0.5 mg) was injected into the right knees (*n* = 12 per condition) of 13-month-old (aged) homozygous LOXL2-overexpressing mice or wild-type littermates (WT). 

Treadmill behavior is useful for preclinical behavioral assessment of chronic pain and inflammation [20]. The treadmill analysis was performed with a standard protocol [32] at 28 days post-MIA injection. The mice were acclimatized to treadmill running (TSE Systems) on 3 consecutive days followed by resting for 1 day before evaluating performance [32]. The acclimatization consisted of a 5-min rest on the treadmill conveyor belt followed by 5 min of running at 7.2 m/s and 5 min at 9.6 m/s. On day 0, the mice were subjected to a graded maximal running test consisting of an initial 5-min rest, after which the running protocol commenced at 4.8 m/min, gradually increasing by 2.4 m/min every 2 min. At all times, the belt was kept at a 5-degree incline. The maximal running speed was defined as the fastest speed at which the mice were able to run for 5 consecutive seconds without touching the electric shock grid at the back of the treadmill. The allodynia was evaluated by Von Frey Hairs test. The researcher conducting the test was blinded to experimental groups.

## Figures and Tables

**Figure 1 ijms-20-04798-f001:**
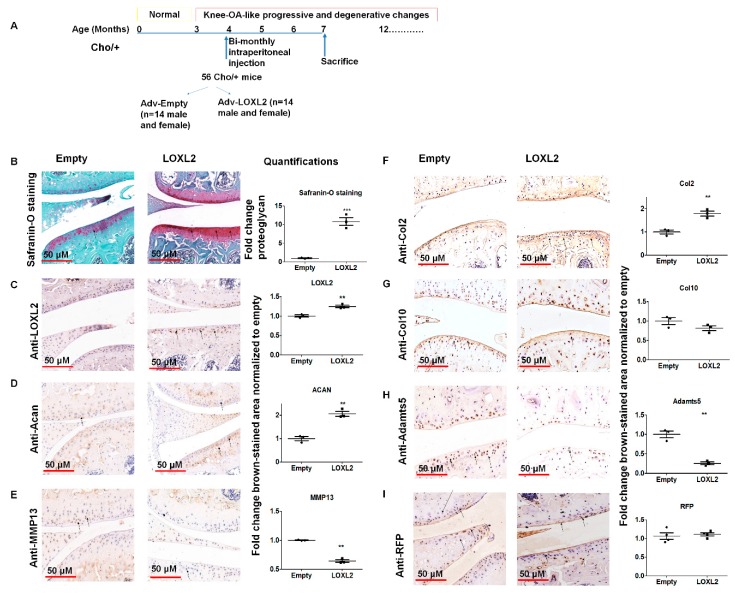
LOXL2 induces a protective response in Cho/+ mice knee joint articular cartilage. (**A**) Scheme of experimental groups and the treatment. (**B**) Safranin-O/Fast green staining of TMJ condylar cartilage in Adv-RFP-Empty compared to Adv-RFP-LOXL2 adenovirus-injected mice, quantification. Immunostaining and quantification of (**C**) LOXL2; (**D**) Acan; (**E**) MMP13; (**F**) Col2; (**G**) Col10; (**H**) Adamts5 and (**I**) RFP in Adv-RFP-Empty and Adv-RFP-LOXL2 treated mice. The statistically significant differences in immunostaining were evaluated by one-way ANOVA with Bonferroni correction (* represent significant differences; ** *p* < 0.01, *** *p* < 0.001; ANOVA).

**Figure 2 ijms-20-04798-f002:**
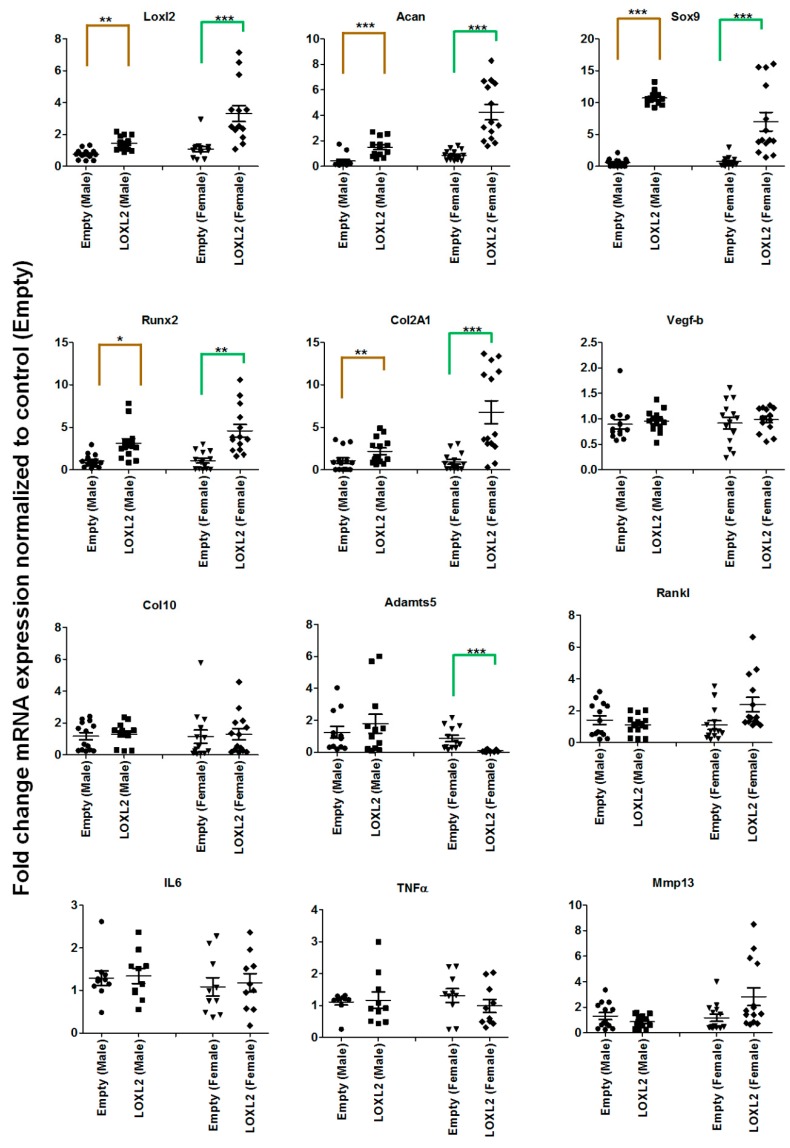
LOXL2 induces mRNA expression of anabolic genes in the Cho/+ mouse model. Each panel shows fold-change in mRNA levels of differentially regulated genes in male and female Cho/+ mice injected with Adv-RFP-Empty (dot-male; triangle- female) and Adv-RFP-LOXL2 (square-male; rhombus-female). The statistically significant differences between groups for males (gold color line) and females (green color line) were evaluated by two-way ANOVA with Bonferroni correction (* represent significant differences; * *p* < 0.05, ** *p* < 0.01, and *** *p* < 0.001; ANOVA).

**Figure 3 ijms-20-04798-f003:**
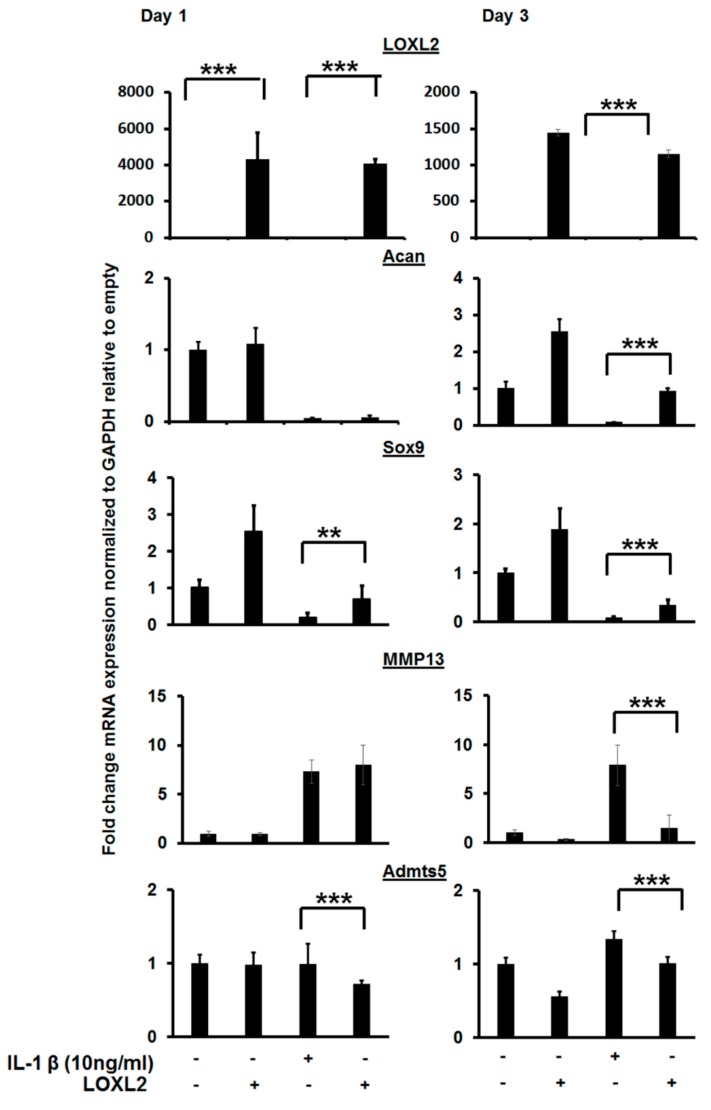
LOXL2 protects against IL-1β-induced effect on cartilage-specific gene expression. RT-qPCR analysis of IL-1β treated ATDC5 cells reduces cartilage-specific gene expression (Acan, Sox9), and LOXL2 protects against this effect. The minus (−) sign represents absence; plus (+) presence of LOXL2 or IL-1β in the respective group. The statistically significant differences in protein or mRNA levels were evaluated by one-way ANOVA with Bonferroni correction (** *p* < 0.01; *** *p* < 0.001; ANOVA).

**Figure 4 ijms-20-04798-f004:**
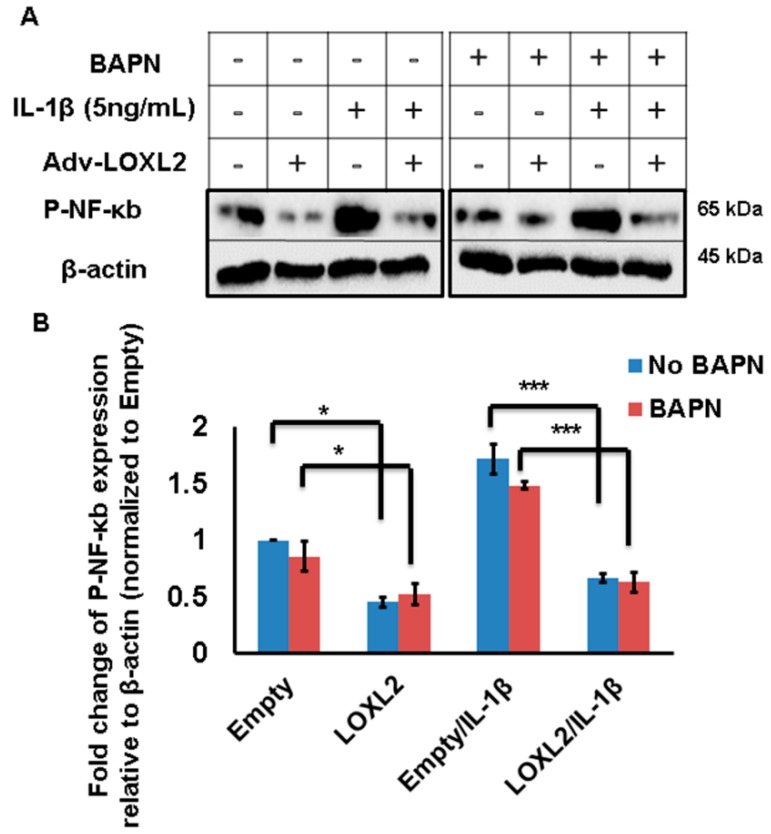
LOXL2 protects against IL-1β-induced NF-κB signaling even in the presence of a LOX family inhibitor. (**A**) IL-1β promotes phospho-NF-κB in ATDC5 cells, and this effect is attenuated by LOXL2 overexpression in the absence or presence of BAPN. The minus (−) sign represents absence wheras plus (+) sign represents presence of LOXL2 or IL-1β in the respective group. (**B**) Quantification of this effect. The statistically significant differences were evaluated by one-way ANOVA with Bonferroni correction for p-NF-κB (* *p* < 0.01, *** *p* < 0.001; ANOVA).

**Figure 5 ijms-20-04798-f005:**
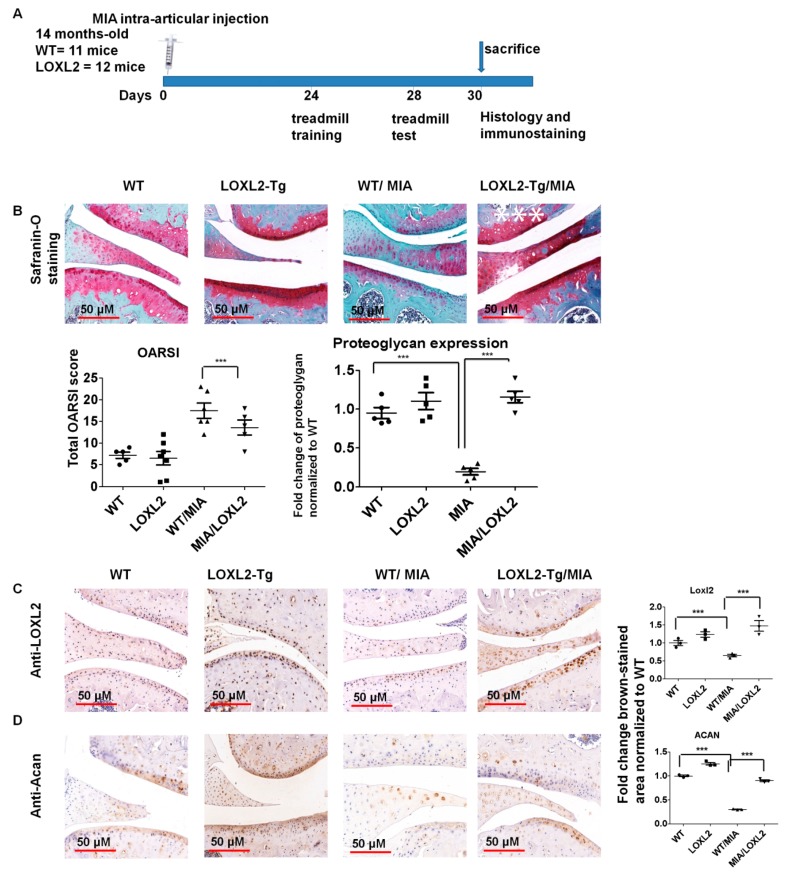
LOXL2 overexpression in transgenic mice protects against MIA-induced OA-related catabolic changes in knee joint articular cartilage. (**A**) Scheme of LOXL2 mice MIA injection groups and its functional analysis; (**B**) Safranin-O staining in the indicated WT and LOXL2 transgenic groups, its quantification and OARSI scoring. Immunostaining and quantification of (**C**) LOXL2 and (**D**) Acan in MIA injected LOXL2 overexpressing or WT mice. The fold change differences in immunostaining for LOXL2 and Acan expression in WT (dot), LOXL2 (square), WT/MIA (triangle) and MIA/LOXL2 (inverted triangles) are shown in adjacent figures. The statistically significant differences in immunostaining were evaluated by one-way ANOVA with Bonferroni correction (*** *p* < 0.001; ANOVA).

**Figure 6 ijms-20-04798-f006:**
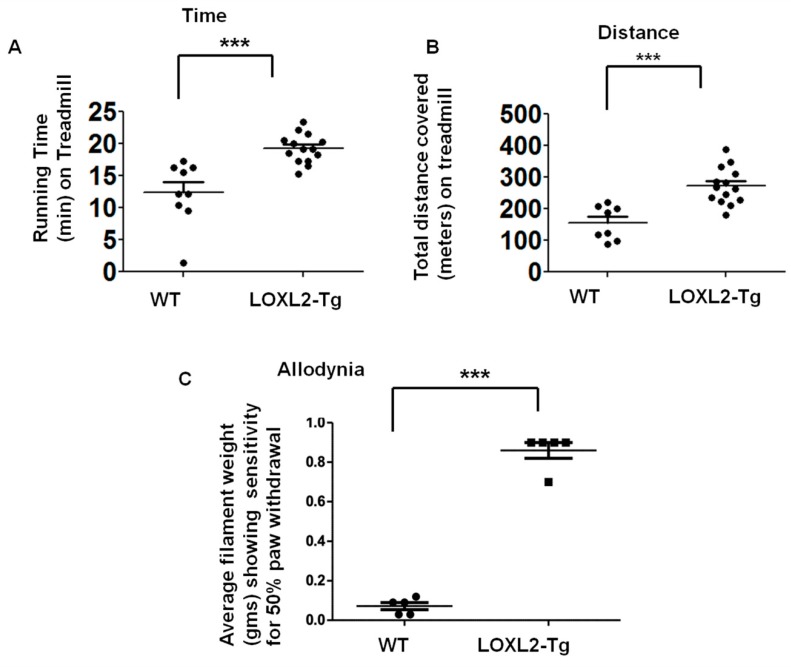
LOXL2 overexpression in transgenic mice protects against MIA-induced OA-related decline in knee joint function. (**A**) Maximal time spent on the treadmill by MIA-induced LOXL2 transgenic and WT mice. (**B**) Total distance covered on the treadmill by MIA-induced LOXL2 transgenic and WT mice. (**C**) Quantification of average Von Frey hairs weight (gms) (as indicated in the graph, *y*-axis), showing pain-sensitive allodynia of MIA-induced LOXL2 transgenic (square) and WT mice (dot). The statistically significant differences were evaluated by one-way ANOVA with Bonferroni correction (*** *p* < 0.001; ANOVA).

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
