# Peer review of "Lysyl Oxidase-Like 2 Protects against Progressive and Aging Related Knee Joint Osteoarthritis in Mice"

_ijms, 2019, doi:10.3390/ijms20194798_

Round 1

Reviewer 1 Report

Lysyl Oxidase-Like 2 Protects against progressive and 3 ageing-related knee Joint Osteoarthritis in mice

Overview: The aim of this study was to determine if adenovirus-delivered LOXL2 protects against progressive knee osteoarthritis (OA). The authors assessed the specific mechanism of LOXL2 action and determined if its overexpression in transgenic mice can protect against the development of OA.

Specific Comments:

The title is appropriate. The abstract is a good summary of the work presented. The keywords are incomplete. Lysyl Oxidase-Like 2 and adenovirus delivery should be added. The data presentation is excellent The conclusions are sound and supported by the data presented. In Figure 4 panel B the shadows from the black horizontal lines should be removed. In summary, this study demonstrates the importance of developing a more detailed understanding of LOXL2 function and that adenovirus LOXL2 is a candidate for future translational applications.

Author Response

Thanks for specific comments and suggestions. Additional keywords Lysyl Oxidase-Like 2 and adenovirus delivery are added. In Figure 4 panel B is corrected.

The title is appropriate. The abstract is a good summary of the work presented. The keywords are incomplete. Lysyl Oxidase-Like 2 and adenovirus delivery should be added. The data presentation is excellent The conclusions are sound and supported by the data presented. In Figure 4 panel B the shadows from the black horizontal lines should be removed. In summary, this study demonstrates the importance of developing a more detailed understanding of LOXL2 function and that adenovirus LOXL2 is a candidate for future translational applications. 

Response: Thanks for specific comments and suggestions. Additional key words Lysyl Oxidase-Like 2 and adenovirus delivery are added. In Figure 4 panel B is corrected.

Reviewer 2 Report

This study examines the benefits of adenovirus-mediated gene transfer of LOXL2 to protect against OA in vivo. The work is well performed and the approach valuable. I have the following points that need critical attention:

the use of adenoviral vecctors for future applications in vivo is a bit problematic: adenoviral gene transfer might induce deleterious responses in such a setup so the authors have to demonstrate that such was not the case in the model selected. Please evaluate therefore potential immune and cytotoxic responses. This should be tested in vivo and cytotoxic assays need to be also performed in vitro. please include an immunodetection of type-II and -X collagens and of relevant ADAMTs (for this one, please also perfgorm a real-time RT-PCR). please provide information on the expression of major inflammatory cytokines in vivo.

Author Response

The use of adenoviral vecctors for future applications in vivo is a bit problematic: adenoviral gene transfer might induce deleterious responses in such a setup so the authors have to demonstrate that such was not the case in the model selected. Please evaluate therefore potential immune and cytotoxic responses. This should be tested in vivo and cytotoxic assays need to be also performed in vitro. Please provide information on the expression of major inflammatory cytokines in vivo.

Response: Thanks for specific comments and suggestions. LDH cytotoxicity data in vitro and in vivo in supplementary figure S1. In addition, the effect on expression cytokines expressed in OA ( IL6 and TNFα) are provided in Figure S1.

Please include an immunodetection of type-II and -X collagens and of relevant ADAMTs (for this one, please also perform a real-time RT-PCR).

Response: Thanks for specific suggestions. Immunostaining for type-II and -X collagens and ADAMTs 5 is included in Figure 1. RT-PCR data for Adamts5, Col2 and 10 are included in Figure 2. The expression of  IL6, TNFα data is also included in Figure 2.

Round 2

Reviewer 2 Report

The revised version does not contain the new figures, als not the supplementary figure(s). I can not therefore make any further recommendation.

Also, correct "adevirus" for "adenovirus" in the new (yellow) text.

many thanks - I can see all is fine now. Please accept on my behalf